# Building an ecological momentary assessment smartphone app for 4- to 10-year-old children: A pilot study

Helen M. Milojevich[1], Daniel Stickel[2], Margaret M. Swingler[3], Xinyi Zhang[4], Jeffery Terrell[5], Margaret A. Sheridan[2], Xianming Tan[4]*

1 Center for Child and Family Policy, Sanford School of Public Policy, Duke University, Durham, North Carolina, United States of America, 2 Department of Psychology and Neuroscience, University of North Carolina at Chapel Hill, Chapel Hill, North Carolina, United States of America, 3 Frank Porter Graham Child Development Institute, University of North Carolina at Chapel Hill, Chapel Hill, North Carolina, United States of America, 4 Department of Biostatistics, University of North Carolina at Chapel Hill, Chapel Hill, North Carolina, United States of America, 5 Department of Computer Science, University of North Carolina at Chapel Hill, Chapel Hill, North Carolina, United States of America

* xianming@email.unc.edu

**Data Availability Statement:** The data is available at https://doi.org/10.5281/zenodo.8220028

## Abstract

### Objective

Ecological momentary assessment (EMA) minimizes recall burden and maximizes ecological validity and has emerged as a valuable tool to characterize individual differences, assess contextual associations, and document temporal associations. However, EMA has yet to be reliably utilized in young children, in part due to concerns about responder reliability and limited compliance. The present study addressed these concerns by building a developmentally appropriate EMA smartphone app and testing the app for feasibility and usability with young children ages 4–10 (N = 20; $m$ age = 7.7, $SD$ = 2.0).

### Methods

To pilot test the app, children completed an 11-item survey about their mood and behavior twice a day for 14 days. Parents also completed brief surveys twice a day to allow for parent-child comparisons of responses. Finally, at the end of the two weeks, parents provided user feedback on the smartphone app.

### Results

Results indicated a high response rate (nearly 90%) across child surveys and high agreement between parents and children ranging from 0.89–0.97.

### Conclusions

Overall, findings suggest that this developmentally appropriate EMA smartphone app is a reliable and valid tool for collecting in-the-moment data from young children outside of a laboratory setting.

**Funding:** This study was supported by NC TraCS grant 550KR221923 (PI: Tan) and NIMH grant R01 MH115004 (PI: Sheridan).

**Competing interests:** The authors have declared that no competing interests exist.

## Introduction

Collecting accurate self-report data, particularly with vulnerable populations such as children and clinical adolescent samples, is a major concern among researchers. One methodology that has gained increased recognition for minimizing recall burden and maximizing ecological validity is ecological momentary assessments, or "EMA", which consists of repeated assessments of behaviors or experiences in real time across a given time frame [1]. EMA has proven to be an incredibly versatile tool to (1) characterize individual differences, (2) describe natural history, (3) assess contextual associations, and (4) document temporal associations across numerous types of sample populations (e.g., married couples, adolescents, medical patients; [1–3]). However, despite the clear methodological strengths, the repeated questioning over lengthy periods outside of a laboratory setting makes utilizing EMA with certain populations, particularly young children, challenging and leads to issues of participant burden, limited compliance, and concerns over the responder reliability [1]. The goal of the present study was to addresses these issues by building a developmentally appropriate EMA smartphone app and testing the app for feasibility and usability with young children 4 to 10 years of age.

Research using EMA has taken numerous methodological forms, with early work relying on paper-and pencil daily diaries, pagers, and palm pilots, while more recent EMA studies utilize smartphone or tablet apps [4]. Most EMA studies include active data collection, which requires participants to respond to questionnaires at several pre-specified or random times each day during a study period. For example, in one EMA study participants were asked to answer 20 questions, 2 times a day, for 14 consecutive days [5]. In another EMA study, participants were asked to answer a survey with 40 questions at up to 7 random times per day for about 45 days [1]. Of note, the timing and frequency of EMA surveys generally fall into one of two categories: time-based sampling and event-based sampling [6]. Time-based sampling refers to fixed or variable survey intervals that are typically initiated by the EMA device (e.g., smartphone app). Within these studies, participants do not control the timing or frequency of the surveys. In contrast, with event-based sampling, surveys are initiated based off events of interest (e.g., consumption of alcohol or self-injurious behaviors), which allows the participant to select when a survey should be completed. Both EMA designs have been used frequently with adult populations, including high-risk and clinical samples [7, 8], as well as with older children and adolescents [5, 9–18].

### EMA with older children and adolescents

Ample evidence suggests that EMA can be used effectively and reliably with older children and adolescents (for review, [6, 19, 20]). For example, in a proof-of-concept study, Suveg and colleagues [15]examined whether EMA was a feasible method of monitoring transitory emotional state with a school-age, community sample of youth (mean age of 9). Overall, results indicated that youths' responses to the EMA diaries significantly correlated to parent-report and self-report of emotional functioning, suggesting the youth were reliably reporting on their daily emotional fluctuations. More recently, EMA has been expanded beyond community samples to include high-risk youth populations, such as those with clinical depression and anxiety [11], attention-deficit/hyperactivity [10, 13, 14, 16, 18, 21], pregnant adolescents [17], and adolescents with self-injurious behaviors [5]. The combined findings across EMA studies indicates that the use of EMA is supported with children and adolescents as young as 8 years of age across gender, racial/ethnic groups, nationality, socioeconomic status, and mental or physical health [19]. Despite considerable support for the use of EMA in older youth, given

methodological and developmental barriers, EMA studies with children younger than 8 may require protocol adaptations.

## Challenges to using EMA with young children

To date, EMA has been almost used exclusively in children 8 years or older, in part due to concerns about young children's ability to understand and adhere to EMA protocols. For one, young children may lack the ability to read survey questions administered during EMA. One potential solution to this barrier recommended by Heron and colleagues [19 is to use pictorial response options or thermometers in place of traditional Likert scales as has been done for decades in other domains of child research [22–25]. Another challenge to utilizing EMA in young children is the concern about providing young children with mobile phones or tablets that could be used for non-study purposes. Solutions may include limiting access to the device, using reduced data plans that do not allow for internet access, or providing the phone to parents for monitoring. Other protocol adaptations to reduce participant burden and increase compliance and reliability include training young children and their caregiver on the EMA device and allowing children to practice responding to surveys, monitoring compliance during data collection, and providing compliance-based incentives [19].

## Present study

The overarching goal of the present study was to build a developmentally appropriate EMA smartphone app and test the app for feasibility and usability with young children 4 to 10 years of age. First, our research team built the child EMA app. Second, we recruited a sample of 20 children ages 4–10 and their parents to pilot test the app and provide feedback on its feasibility and usability. During pilot testing we anticipated a relatively high survey response rate of approximately 75% based on previous EMA studies with children [19, 20]. We also predicted a significant correlation between child and parent responses [15]. Finally, we explored the range of children's responses, parents' perceptions of the app, and predictors of children's responses (e.g., age, gender).

## Method

### Participants

Participants included 20 young children ages 4–10 ($m$ = 7.7, $SD$ = 2.0) and their parents ($n$ = 20; 90% mothers). Families were recruited from a database of families who had previously participated in studies conducted by the research team. Families from the database were recruited if their child was between 4 and 10 years of age and if the family spoke English well enough to consent and complete study procedures. Exclusion criteria were based on the criteria used for the creation of the database and included major medical conditions, neurological illness, pervasive developmental disorders, or prenatal substance exposure. According to their parents, 70% of the children identified as girls; approximately 40% of the children identified as white, 47% as Black/African American, and 5% as Asian; additionally, 10% identified as Hispanic/Latinx (5% did not report on race or ethnicity). Over half of the parents were married (57.1%) and 68.5% had a least some college education. Finally, 40.0% of parents were employed part-time and 12.0% were unemployed (looking for work or currently a student). One additional family was excluded due to technical difficulties (not able to turn off the app appropriately) with the app.

## Development of the child EMA app

The design of the EMA app and the testing procedures were informed by previous meta-analyses and systematic reviews of the use of mobile EMA with children and adolescents [19, 20]. The EMA child app was designed for both android and iOS operating systems and was made available for parents to download onto their existing smartphones, thereby removing the need to carry a study-specific phone. The EMA child app administered twice daily surveys at predetermined times of day in the morning and evening. These time frames were selected as they were the times of day when parents were likely to be with their children (e.g., before and after school/work). For each survey, parents were allowed to select a two-hour time window to have their child complete the questions. For example, some parents selected to have their child complete the morning survey between 6-8am, while others opted for 7-9am or 8-10am. For each survey, the app included 11 questions (S1 Appendix), which were presented sequentially, each on its own page. When a question page appeared on the screen there was a star-shaped cartoon figure that said the survey question aloud, which removed the need for the children to read the survey questions. The questions also appeared in speech bubbles next to the cartoon character (S2 Appendix). Children responded either Yes/No using thumbs up/thumbs down images or using visual Likert scales (S1 Appendix) based on previous cross-sectional and longitudinal studies with children of this same age range [22, 25]. After completing the survey questions, the app presented a prize page showing the children how many points they had earned for completing the survey.

The prize page was included based on feedback from other researchers who had conducted EMA studies with older children [10]. These researchers noted that children tended to respond regularly and reliably to EMA survey questions when they were happy or generally in a good mood; however, when children were feeling more negatively (e.g., angry, sad) their response rates dropped. As such, it was recommended that we incentivize children to participate by tying their compensation to the number of completed surveys as has been done in EMA studies with children and adults [19]. In the present study, children earned 5 points for each completed survey and the number of total points earned corresponded to receiving a small (0–62 points worth $25), medium (63–123 points worth $50), or large (124–200 points worth $70) prize. At the start of their participation, children were shown the prize options and asked to select one medium and one large prize that they wanted to try to earn. For example, a child might select a Lego castle (large prize) at the start of the study, then as they complete the surveys each day the prize page in the app depicted how close the child was to earning the castle. It was decided before the beginning of the study that all children would receive at least the medium prize regardless of performance (although children were not told this until after participation). All but one child received the large prize that they had selected. Families were also offered the option of a gift card of the same monetary value as the prize earned (e.g., $70 for a large prize) in place of a prize in case the child did not like any of the prize options.

Participants' responses, appropriately coded, were automatically sent to a pre-specified virtual server and were then downloaded to a campus computer. This allowed the researchers to monitor compliance in real-time. No identifying information was involved in data transmission.

## Procedure

Families who agreed to participate were scheduled for a 30-minute study session at their homes. During the study session, a research assistant reviewed the study information and obtained written informed consent from the parent, which was then duly documented and witnessed by the research team. Following consent, the research assistant introduced the EMA

child app and how to interact with the smartphone app. Parents were then instructed in how to download the app to their existing smartphone (45% Android vs 55% iOS). After installation, the child and parent spent several minutes interacting with the app to learn its functionality and how to respond to survey questions. Once the child and parent felt sufficiently comfortable with the app, the research assistant reminded the family of the survey schedule (twice a day for 14 days) and then provided compensation for the study visit.

During the 14-day EMA study, children completed 2 assessments a day that asked developmentally appropriate questions about their mood and behavior. When the time window opened for a given survey, parents were notified by an alert from the app and were instructed to give the phone to their child so that the child could complete the survey questions. Parents were instructed and trained to allow their child to complete the survey questions by themselves without help from or supervision by the parent. After the child completed their survey, parents were also asked to complete a subset of the survey questions, namely 5 questions about the child's behavior and whether the child was at home, to allow for comparisons in responses across child and parent.

We also incorporated an adaptive feature in this EMA study (see Blinded for Review). Specifically, for the child EMA surveys, we tried to predict each participant's responses to each question based on past responses from all subjects. Based on the performance of the predictions, we identified for each participant survey questions in each survey whose responses could be skipped, i.e., could already be accurately predicted based on past responses. Thus, the surveys become adaptive in that participants could receive a different list of survey questions across time and across participants. We turned off this adaptive feature for 5 (out of 20) participants so that we could examine whether participants responded differently with this adaptive feature versus without it. No differences emerged for participants with or without the adaptive feature. The present study was part of a larger project to build and test intelligent ecological momentary assessment (iEMA). The creation and testing of the iEMA adaptive feature are described in detail in Blinded for Review. Findings from the larger study suggest that iEMA holds the potential to upgrade current practice of conducting EMA studies, in that it collects needed information at needed times, continuously alleviates the burden of assessments on participants, and hence improves validity, plausibility, and cost-effectiveness of most EMA studies. In the present study we included the adaptive feature for 5 of the participants as a pilot test of the iEMA algorithms in children. Although we found no differences in responses for participants with or without this adaptive feature, additional research with larger samples is needed to confirm validity and refine the iEMA algorithms, particularly with child samples.

On the final day of assessments, parents also completed an ease of use and satisfaction survey about the EMA app experience. Active survey responses were continuously tracked and stored in a university encrypted data server. If a family failed to complete surveys for two days in a row, they were contacted by the research team and given a reminder to complete as many surveys as possible. If the parent reported technical difficulties the research assistant troubleshooted these issues and helped the family resume the EMA surveys. Following the 2-week EMA study, parents were compensated for their participation and children received a prize. Each family received an $80 gift card as well as the medium or large prize that the child had selected at the first visit.

## Statistical analyses

Data were analyzed in SPSS 27.0 IBM Statistics, IBM Corp., Armonk, NY). First, preliminary analyses were conducted to examine the percentages of responses on the child surveys and differences in response rates across days comparing weekends to weekdays. Next, we examined

children's response options for each of the survey questions to determine variations in responses across children and questions (e.g., mad vs. happy). We then calculated the percentage of agreement between children's responses and parents' responses to each of the 5 behavioral questions to determine agreement rates. Finally, we explored the user feedback provided by parents.

## Results

Preliminary analyses examined the percentages of responses on the child surveys. The average valid response rate across child participants was 89.1% (range = 64% to 100%); i.e., children received the app alert and completed the survey as expected. Approximately 7% percent of responses were considered "participant error", i.e., despite receiving the app alert the child did not complete the survey. Finally, 3.75% percent of responses were classified as "technical error", in which case the survey was not completed due to a technical issue with the smartphone app (e.g., trouble getting app to open).

We examined differences in response rates across days (i.e., the total number of prompts out of 2 that were answered). Results indicated that there were similar valid response rates on weekends (Saturday and Sunday; $M = 1.72$, $SD = 0.46$) compared to other weekdays ($M = 1.80$, $SD = 0.51$; $t = -1.24$, $p = 0.22$). A $t$-test is a statistical test that is used to compare the means of two groups. Larger $t$ scores reflect more difference between the groups [26]. A $p$-value of less than .05 on a $t$-test indicates a statistically significant difference between means [27].

### Survey responses

For each of the 11 survey questions, children used the full range of response options (see S3 Appendix for within person variations in responding). For question 1, 80.81% of children indicated that they were at home at the time of the survey, while 19.19% reported being outside of the home. Results of the emotion survey questions are reported in Table 1. The most frequent response for Happy was 5 (very happy, 45.9%), for Mad was 1 (not mad, 73.1%), for Sad was 1 (not sad, 70.7%), for Excited was 5 (very excited, 35.3%), for Tired was 1 (not tired, 40.5%). Regarding the behavioral questions (Table 2), children reported getting in a fight with someone in their family on 10.4% of the surveys, indicated getting in trouble at school on 2.2%, yelled at someone on 8.0%, hit or kicked someone on 2.8%, and had something bad happen on 7.62% of the surveys. Of note, all children reported at least one negative experience over the course of the 14-day survey period. These results suggest that although the base rate for these behaviors or experiences was reportedly low across the surveys, children did still endorse them on occasion.

### Correspondence between child and parent responses

We calculated the percentage of agreement between children's responses and parents' responses to each of the 5 behavioral questions and found that the agreement rate was overall high (Table 3), ranging from 0.89–0.97, with the highest agreement rate for Q1 (at home or not) and Q10 (Hit/kick someone), and the lowest agreement rate for Q7 (fight with someone). The percent agreements are reported using the correlation coefficient $r$, which is the statistical measure of the strength of a linear relationship between two variables. To interpret, $r$ values range from -1 (perfect negative linear relationship) to +1 (perfect positive linear relationship) [28].

**Table 1. Child responses to emotion survey questions.**

|  | Happy | Mad | Sad | Excited | Tired |
|---|---|---|---|---|---|
| 1 (Neutral) | 8.8% | 73.2% | 70.7% | 22.4% | 40.5% |
| 2 | 9.0% | 6.6% | 10.8% | 15.8% | 20.0% |
| 3 | 20.4% | 2.2% | 5.0% | 12.6% | 14.0% |
| 4 | 15.6% | 3.8% | 2.8% | 13.6% | 8.6% |
| 5 (Extremely) | 45.9% | 5.8% | 8.4% | 35.3% | 15.0% |
| Mean (SD) | 3.8 (1.3) | 1.5 (1.1) | 1.6 (1.2) | 3.2 (1.6) | 2.4 (1.5) |
| Missing | 0% | 2.2% | 0.6% | 0% | 1.8% |
| Skipped* | 0.2% | 6.2% | 1.6% | 0.2% | 0% |

*Note.* Skipped indicates that the questions were skipped by the EMA algorithm. Missing refers to questions that were asked, but not answered.

## User feedback from parents

Parents' responses to the post-study survey showed that overall parents had positive perceptions of the child app (see Table 4). Specifically, 90.5% of the parents indicated that the use of the app for the study did not interfere with their regular life, 95% of them did not consider it hard to comply with the study protocol, and 100% of them would participate in another study like this. Per app features, 76% of them consider the reading aloud feature of the app as helpful for their child and 81% considered the visual features as useful to improve engagement. The EMA's adaptive feature was well accepted, as only 5% of them indicated that this adaptive feature made it difficult to answer questions.

Parents were also asked to provide any additional feedback that was not covered in the feedback survey. Five of the 20 parents provided additional feedback. Some of the parents of older children (i.e., 8–10-year-olds) noted that the audio on the app was too slow for their children who could read the text faster than the app's audio. Another parent commented that the term "fight" was ambiguous in the question "So far today, did you fight with someone in your family". One parent reported that their child seemed concerned about disclosing negative experiences and wanted to "say the right thing to make sure everything seemed okay". Overall, most parents commented that their child seemed to enjoy interacting with the app and they felt their child responded accurately to the surveys.

For the 15 subjects with the adaptive feature on, based on the machine learning algorithms, some questions in surveys were skipped. The total number of questions (out of 308 = 14*2*11) skipped for these 15 participants ranged from 12 to 46, with a mean of 31 ($SD$ = 10.2). The frequency of skipping an emotion survey question ranges from 0.3% (Q6 Tired) to 8.5% (Q3: Mad), see Table 1. That is, in about 8.5% of the surveys, we skipped the question about Mad. The frequency of skipping a behavior survey question ranges from 14% (Q7 Fight) to 21% (Q10: Hit/kick, Q8: get in trouble), see Table 2. Our algorithm tended to skip questions with smaller variation in the responses more often, as shown in Tables 1 and 2. The missing rate, response distribution, and agreement between parent and child responses for each survey question was similar for the 15 children with adaptive feature on vs. for the 5 children with the

**Table 2. Child responses to behavioral survey.**

|  | Q1. Home Right Now? | Q7. Fight with Someone? | Q8. Get in Trouble? | Q9. Yell at Someone? | Q10. Hit/kick Someone? | Q11. Something Bad Happen? |
|---|---|---|---|---|---|---|
| Yes | 71.3% | 10.4% | 2.2% | 8.0% | 2.8% | 7.6% |
| Missing | 0.6% | 1.0% | 1.2% | 1.4% | 1.4% | 1.6% |
| Skipped | 11.0% | 10.4% | 15.4% | 13.2% | 15.4% | 12.6% |

**Table 3. Percent agreement between child and parent response.**

| Q1. Home Right Now? | Q7. Fight with Someone? | Q8. Get in Trouble? | Q9. Yell at Someone? | Q10. Hit/kick Someone? | Q11. Something Bad Happen? |
|---|---|---|---|---|---|
| .97 | .89 | .90 | .90 | .96 | .93 |

adaptive feature off. This suggests that the adaptive feature reduced the number of questions requiring responses, without losing information or changing response patterns among participants.

## Discussion

The present study sought to develop and test a smartphone app designed to capture EMA data from young children reliably and with validity. Overall, our findings suggest that by using a smartphone app that was designed to be developmentally appropriate, EMA methodologies can be implemented even in children as young as 4 years of age. In this study we were able to attain high compliance and reliable use of the app for children. This was in part because we applied classic behavioral modification/learning theory to the administration of the app [29]. That is, instead of opting for motivating children to comply with app use via social contract (e.g., convincing them about the scientific importance of the app) or use of punishment (e.g., deducting points if the child did not comply) we used positive reinforcement. Positive reinforcement consistently elicits the most compliant behavior in general and, in particular, for children [30]. An additional factor is that many of these children were part of an ongoing longitudinal study. They and their parents had significant levels of interaction with the study staff which likely led them to be comfortable with new procedures explained by familiar staff members. Below we discuss our findings as well as present some lessons learned and best practices for future research and implementation.

First, in line with our hypotheses, we obtained a high survey response rate of over 80% across the twice daily surveys. Moreover, parent and child responses were significantly correlated, suggesting high levels of agreement between the two responders. Importantly, we asked participants about a range of children's emotions and behaviors and found that children seemed willing and able to report on their emotions and even disclose negative behaviors (e.g., fighting). Responses on emotions spanned the full scale (e.g., from neutral to extremely happy) indicating that children did not rely solely on the extreme ends of the emotion scales, but rather responded with variation and some nuance in the intensity of their emotional states. These findings map onto results with older children and adolescents [6, 19, 20].

**Table 4. Post-study parental feedback survey.**

| | Probably True | Not Sure | Probably False | No Opinion | Missing |
|---|---|---|---|---|---|
| 1. The study interfered with my regular life | 9.5% | 0% | 90.5% | 0% | 0% |
| 2. It was hard to comply with the survey results | 0% | 4.8% | 95.2% | 0% | 0% |
| 3. Reading questions aloud to my child was helpful | 76.2% | 9.5% | 14.3% | 0% | 0% |
| 4. The app visuals helped my child engage in the questions | 80.9% | 9.5% | 0% | 9.6% | 0% |
| 5. Allowing delayed responses to surveys was helpful | 35% | 0% | 0% | 65% | 0% |
| 6. Reminders to complete a missed survey was helpful | 50% | 0% | 0% | 50% | 0% |
| 7. The adaptive feature—changing orders of questions—made answering questions more difficult | 4.8% | 4.8% | 52.3% | 38.1% | 0% |
| 8. The prizes helped my child want to complete the surveys | 95.2% | 4.8% | 0% | 0% | 0% |
| 9. We would participate in another study like this. | 100% | 0% | 0% | 0% | 0% |

Second, several best practices emerged from the present study. Regarding the design of the smartphone app, parents reported that the visuals of the app (e.g., the star character and picture response options) helped their children engage in responding to the questions. Similarly, parents reported that having the questions read aloud via the app was crucial for pre-reading children. This feature also allowed children to complete the surveys by themselves without the aid of their parents. Another best practice from the present study is training. Specifically, training both child and parent participants to the EMA smartphone app and study procedures allowed participants to troubleshoot any technical issues and gain familiarity with the app prior to the start of the surveys. This study was conducted during the COVID-19 pandemic and, therefore, at times in-home study visits were not feasible given local health orders. However, training was conducted for all participants via porch drop-offs and outside visits, thereby allowing in-person training to continue. Parents reported that these trainings were very beneficial, especially for helping their child gain confidence and competence with the app.

Finally, the present study utilized several strategies for enhancing participant compliance, including compliance monitoring, and providing compliance-based incentives. Compliance monitoring was conducted throughout the study, such that when children missed two surveys in a row, our research team contacted the family to provide reminders about the surveys and troubleshoot any issues. Parents reported that these reminders supported the high response rate obtained in the current study. Additionally, as described in the Method, the present study included compliance-based incentives to encourage children to complete the surveys. The key features of this reward system were to allow the child participant to select one large and one medium prize at the start of the study that they could "earn" by completing surveys. After completing each survey, the child was shown a prize page within the app that included a thermometer picture display of how many points they had earned and how close they were to earning the medium and large prizes. Importantly, children were given at least the medium prize regardless of their survey completion rate (all but one child earned the large prize) to ensure that they enjoyed participating in the study, however, during the study children believed that their prize level was contingent on compliance with the study procedures. At the conclusion of the study, parents reported that the prizes helped incentivize their children to complete the surveys.

## Limitations and future directions

The present study included the creation and testing of an EMA smartphone app in young children, allowing researchers for the first time to assess children's emotions and behaviors outside of the laboratory and through children's own reports. Despite these strengths, the present study has limitations that should be addressed in future research. First, we included a range of young children (ages 4–10, $m = 7$) as participants, including some that overlap with ages previously included in EMA studies (8–10 years). However, future studies would benefit from testing the smartphone app in an exclusively early childhood sample (4–7 years) to ensure that findings on the utility and validity of the app remain even in the youngest children. Second, as with many pilot studies, the included sample size was relatively small, therefore future research should test the EMA smartphone app in a larger sample of children, preferably with a mix of demographic characteristics (e.g., urbanicity, socioeconomic status, adversity exposure) to determine generalizability. Finally, the present study was conducted during the COVID-19 pandemic. Public health restrictions and disruptions caused by the pandemic may have influenced participant behavior and engagement. As such, future research should confirm the findings of the present study now that the emergency phase of the pandemic has ended.

The pathways for future research are numerous. One pathway that holds promise is the use of the smartphone app for interventions or treatments. This intervention approach, referred to as ecological momentary intervention (EMI; [31]), has gained considerable momentum with adult and adolescent populations, with very little use in pediatric samples. Of note, the flexibility of the EMA methodology allows EMI to tailor content to individuals and alter the timing of specific intervention components. As such, interventions and treatments can be delivered in real-time when individuals are most in need of intervention. Future research should explore the feasibility and efficacy of integrating intervention or treatment components into the EMA app, and subsequently evaluate its impact on children's behaviors.

## Conclusion

The present study created and tested an EMA smartphone app in young children. The development of the smartphone app was informed by the extant EMA literature with older children and adolescents and highlights best practices for future research. Overall, findings suggest that the app was well received by both parents and children. Response rates were high and response agreement between parents and children was impressive. Future researchers should utilize the lessons learned in the current study to expand the use of EMA and EMI in young children.

## Supporting information

**S1 Appendix. This lists all EMA survey questions and response options.**
(DOCX)

**S2 Appendix. The cartoon character and speech bubbles that appear in the EMA app.**
(DOCX)

**S3 Appendix. Change of response (Happiness and Madness) over time for 5 selected participants.**
(DOCX)

**S4 Appendix. Predictors of child responses.**
(DOCX)

## Acknowledgments

The authors thank the two anonymous referees for their constructive comments which helped us to improve our manuscript.

## Author Contributions

**Conceptualization:** Helen M. Milojevich, Margaret M. Swingler, Jeffery Terrell, Margaret A. Sheridan, Xianming Tan.

**Data curation:** Daniel Stickel, Xinyi Zhang.

**Formal analysis:** Helen M. Milojevich, Xinyi Zhang, Xianming Tan.

**Funding acquisition:** Margaret A. Sheridan, Xianming Tan.

**Investigation:** Helen M. Milojevich.

**Methodology:** Helen M. Milojevich, Daniel Stickel, Margaret M. Swingler, Xinyi Zhang, Jeffery Terrell, Margaret A. Sheridan, Xianming Tan.

**Project administration:** Helen M. Milojevich, Daniel Stickel, Xinyi Zhang, Margaret A. Sheridan, Xianming Tan.

**Resources:** Jeffery Terrell, Margaret A. Sheridan, Xianming Tan.

**Supervision:** Helen M. Milojevich, Margaret A. Sheridan, Xianming Tan.

**Writing – original draft:** Helen M. Milojevich.

**Writing – review & editing:** Daniel Stickel, Margaret M. Swingler, Margaret A. Sheridan, Xianming Tan.

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
