## [Decision Letter · Decision Letter 0]

9 Jun 2023

PONE-D-23-04757Building an Ecological Momentary Assessment Smartphone App for Young Children: A Pilot StudyPLOS ONE

Dear Dr. Tan,

Thank you for submitting your manuscript to PLOS ONE. I do want to personally apologize for the delay in returning the results of the review. I had a very difficult time securing two reviewers. After careful consideration, we feel that it has merit but does not fully meet PLOS ONE’s publication criteria as it currently stands. Therefore, we invite you to submit a revised version of the manuscript that addresses the points raised during the review process.

The topic of your paper is valuable and critical. However, both reviewers made important suggestions that will strengthen the paper markedly. I concur with their suggestions. Please carefully consider all of their comments, with a particular focus on incorporating additional information on methods and interpreting the results.

We look forward to receiving your revised manuscript.

Kind regards,

Yu-Wei Ryan Chen, PhD

Academic Editor

PLOS ONE

Journal Requirements:

Reviewers' comments:

Reviewer's Responses to Questions

**Comments to the Author**

1. Is the manuscript technically sound, and do the data support the conclusions?

Reviewer #1: Yes

Reviewer #2: Yes

2. Has the statistical analysis been performed appropriately and rigorously? 

Reviewer #1: Yes

Reviewer #2: Yes

3. Have the authors made all data underlying the findings in their manuscript fully available?

Reviewer #1: Yes

Reviewer #2: Yes

4. Is the manuscript presented in an intelligible fashion and written in standard English?

Reviewer #1: Yes

Reviewer #2: Yes

5. Review Comments to the Author

Reviewer #1: It was a pleasure to read this article. The developed app which allows young children to participate in EMA studies bears great potential to answer novel research questions about the daily life of children. I applaud the authors who took a lot of effort to adapt an app to the needs of young children (rewards, reading out questions, visual scales etc). I am looking forward to read articles which use this app.

I have only some minor comments.

First, I would consider removing the section "Predictors of Child Responses". As you have only 20 participants it is quite difficult to make reliable predictions. If you do not find any differences (as you did) it is probably due to low power (see: r =.40 did not reach significance), and if you had found differences it could be due to one outlier. So I think that your data is not fit to make such inferences. It could be interesting to present descriptives in an Appendix.

Second, I am wondering if you could provide more information about the post-study feedback survey. Was there more questions beside the 9 Likert-scale questions? I would be interested to hear if you have any qualitative data or experiences in contact during monitoring/technical trouble shooting. What kind of problems occurred? What are the lessons learned for a next study with this app? Had the parents ideas how to improve the app? This could be valuable information for readers who are developing an app or working with young children. (In my experiences with EMA studies in adolescents, there is always something that isn't going according to plan.)

Third, I have some questions about your "adaptive feature". I understand that you do not explain this feature at length as this is not the main focus of the article, and probably there is more information in the blinded external resource. If I understand this feature correctly the algorithm for example sees a pattern that if previously children said that their are "mad" and "had a fight", that they probably also would say that they "got in trouble" and therefore this item is not displayed, right? It is a nice idea that we can reduce participant burden with such algorithms, but I am wondering if there is really enough data to make such decisions. I would assume that life is much more diverse than that we can reliably predict answers after less than two weeks (with max 28 datapoints per person). Such variation in life is the reason we do EMA research after all. So can we really be sure that no information is lost. To come to the conclusion that no information is lost would require to check what the algorithm would predict and then check if participants really gave this answer. Please feel free to correct me if I misunderstood this feature or your conclusions.

I am looking forward to your response,

All the best,

Anne Bülow

Reviewer #2: The present study developed and tested a developmentally appropriate EMA smartphone app to assess mood and behavior in young children (ages 4-10), demonstrating high response rates and agreement between parents and children, indicating the app's reliability and validity for collecting in-the-moment data outside of a laboratory setting. This is a well-written study that could inform future EMA studies in this population. I provided some suggestions below:

1. I recommend that the authors reconsider the use of 'young' in their title. When reading the title, I initially expected the study to focus on preschool-aged children. However, since the study included children across early to middle childhood, I believe that 'young children' might not accurately describe the entire age range of the study sample. It might be more appropriate for the authors to include the specific age range in the title to provide a clearer representation of the participants.

2. What specific behavioral theories or frameworks guided this study? It would be helpful if the authors could present the theories or frameworks that influenced factors such as high compliance, the child's confidence and competence with the app, and the concept of providing prizes for the child's participation.

3. The following sentence is counterintuitive. Please separate employed from unemployed. “Finally, 52.6% of parents were either employed part-time or were unemployed (looking for work or currently a student).”

4. Please include the inclusion and exclusion criteria used for this study.

5. Please provide the costs for the gifts that the children received, and the dollar amount offered for the gift card option.

6. Please add a statistical analysis section before the Results section.

7. Define ts, rs and ps for the reader.

8. In the discussion, the authors noted that this pilot study was conducted during the COVID-19 pandemic. Please add this information in the methods section, participants’ subsection. Additionally, it would be helpful to specify the location (city, state) where the study was conducted. Lastly, it is important to acknowledge that the COVID-19 pandemic may have influenced participant behavior and engagement. The restrictions and disruptions caused by the pandemic could have had an impact on the participants' behaviors, which should be acknowledged as a limitation of the study.

9. In the final discussion paragraph about future research, it would be beneficial for the authors to reframe the last sentence. While the authors acknowledged the potential use of the app for interventions or treatments (EMI), they did not provide empirical evidence of its effectiveness in this regard. Therefore, I suggest reframing it as a recommendation for future research to focus on exploring the feasibility and efficacy of integrating intervention or treatment components into the app, and subsequently evaluating its impact on children's behaviors.

6. PLOS authors have the option to publish the peer review history of their article (what does this mean?). If published, this will include your full peer review and any attached files.

Reviewer #1: **Yes: **Anne Bülow

Reviewer #2: No

---

## [Author Response · Author response to Decision Letter 0]

19 Jul 2023

Our point-to-point responses are included in the file "PLOS ONE_Reviewer Response Letter.docx"

---

## [Decision Letter · Decision Letter 1]

3 Aug 2023

Building an Ecological Momentary Assessment Smartphone App for 4- to 10-Year-Old Children: A Pilot Study

PONE-D-23-04757R1

Dear Dr. Tan,

We’re pleased to inform you that your manuscript has been judged scientifically suitable for publication and will be formally accepted for publication once it meets all outstanding technical requirements.

Kind regards,

Yu-Wei Ryan Chen, PhD

Academic Editor

PLOS ONE

Additional Editor Comments (optional):

Reviewers' comments:

Reviewer's Responses to Questions

**Comments to the Author**

1. If the authors have adequately addressed your comments raised in a previous round of review and you feel that this manuscript is now acceptable for publication, you may indicate that here to bypass the “Comments to the Author” section, enter your conflict of interest statement in the “Confidential to Editor” section, and submit your "Accept" recommendation.

Reviewer #1: All comments have been addressed

Reviewer #2: All comments have been addressed

2. Is the manuscript technically sound, and do the data support the conclusions?

Reviewer #1: Yes

Reviewer #2: Yes

3. Has the statistical analysis been performed appropriately and rigorously? 

Reviewer #1: Yes

Reviewer #2: Yes

4. Have the authors made all data underlying the findings in their manuscript fully available?

Reviewer #1: (No Response)

Reviewer #2: No

5. Is the manuscript presented in an intelligible fashion and written in standard English?

Reviewer #1: Yes

Reviewer #2: Yes

6. Review Comments to the Author

Reviewer #1: It was a pleasure to read the revised manuscript. All my points were adequately addressed.

A tiny last remark: I would expect the information about the statistical analysis (t-test, Pearson correlation, chosen alpha value) in the section “Statistical Analysis” instead of in a footnote in the result section. But it is up to the other reviewer who asked for this extra information, the editor and the authors to decide if they agree with me and make this small change in the final manuscript.

Reviewer #2: I thank the authors for addressing all my comments, and I look forward to seeing this paper published.

7. PLOS authors have the option to publish the peer review history of their article (what does this mean?). If published, this will include your full peer review and any attached files.

Reviewer #1: No

Reviewer #2: No

---

## [Editor Report · Acceptance letter]

21 Aug 2023

PONE-D-23-04757R1 

Building an Ecological Momentary Assessment Smartphone App for 4- to 10-Year-Old Children: A Pilot Study 

Dear Dr. Tan:

I'm pleased to inform you that your manuscript has been deemed suitable for publication in PLOS ONE. Congratulations! Your manuscript is now with our production department. 

Kind regards, 

on behalf of

Dr. Yu-Wei Ryan Chen 

Academic Editor

PLOS ONE